# Prevalence and Antibiotic Susceptibility Trends of Selected *Enterobacteriaceae*, *Enterococci*, and *Candida albicans* in the Subgingival Microbiota of German Periodontitis Patients: A Retrospective Surveillance Study

**DOI:** 10.3390/antibiotics11030385

**Published:** 2022-03-14

**Authors:** Karin Jepsen, Wolfgang Falk, Friederike Brune, Raluca Cosgarea, Rolf Fimmers, Isabelle Bekeredjian-Ding, Søren Jepsen

**Affiliations:** 1Department of Periodontology, Operative and Preventive Dentistry, University Hospital Bonn, Welschnonnenstrasse 17, 53111 Bonn, Germany; friederike.brune@web.de (F.B.); raluca.cosgarea@ukbonn.de (R.C.); sjepsen@uni-bonn.de (S.J.); 2Service Laboratory, Center for Oral & Dental Microbiology, 24103 Kiel, Germany; wolfgang.falk@web.de; 3Clinic for Periodontology and Peri-Implant Diseases, Philipps University Marburg, 35039 Marburg, Germany; 4Clinic of Prosthodontics, Iuliu Hatieganu University Cluj-Napoca, 40006 Cluj-Napoca, Romania; 5Institute for Medical Biometry, Informatics and Epidemiology, University of Bonn, 53127 Bonn, Germany; fimmers@ifsq.de; 6Division of Microbiology, Paul-Ehrlich-Institut, 63225 Langen, Germany; isabelle.bekeredjian-ding@pei.de; 7Institute of Medical Microbiology, Immunology and Parasitology, University of Bonn, 53127 Bonn, Germany

**Keywords:** antibiotics, antibiotic resistance, candida, *Enterobacteria*, *Enterococci*, *Klebsiella*, *Serratia*, periodontitis, periodontal pocket

## Abstract

The periodontal microbiota is ecologically diverse and may facilitate colonization by bacteria of enteric origin (*Enterobacteriaceae*, *Enterococci*) and co-infections with *Candida albicans*, possibly producing subgingival biofilms with high antimicrobial tolerance. This retrospective surveillance study followed periodontitis-associated superinfection profiles in a large patient sample. From 2008 to 2015, biofilm samples from deep periodontal pockets were collected from a total of 16,612 German adults diagnosed with periodontitis. The presence of selected *Enterobacteriaceae*, *Enterococci*, and *Candida albicans* was confirmed in overnight cultures. Antimicrobial susceptibility of these clinical isolates was tested by disk diffusion with antibiotics routinely used for treatment of oral infections, e.g., amoxicillin (AML), amoxicillin/clavulanic acid (AMC), doxycycline (DO), and ciprofloxacin (CIP). The mean annual prevalence of patients harboring *Enterobacteriaceae* in periodontal plaques was 11.5% in total and ranged from 2.5% for *Enterobacter cloacae* to 3.6% for *Klebsiella oxytoca*, 1.1% for *Klebsiella pneumoniae*, 2.8% for *Serratia marcescens*, and 1.5% for *Serratia liquefaciens*. In comparison, the mean detection rates for microbiota typically found in the oral cavity were higher, e.g., 5.6% for *Enterococcus* spp. and 21.8% for *Candida albicans*. Among the *Enterobacteriaceae*, species harboring intrinsic resistance to AML (*Enterobacter* spp., *Klebsiella* spp., *Serratia* spp.) were predominant. Non-susceptibility to AMC was observed for *Serratia* spp. and *Enterobacter cloacae*. By contrast, *Enterococcus* spp. only showed non-susceptibility to DO and CIP. Trends for increasing resistance were found to AML in *Serratia liquefaciens* and to DO in *Enterococcus* spp. Trend analysis showed decreasing resistance to AMC in *Serratia liquefaciens* and *Klebsiella oxytoca*; and to DO in *Serratia marcescens, liquefaciens*, and *Enterobacter cloacae*. This study confirms the low but consistent presence of *Enterobacteriaceae* and *Enterococci* among the subgingival microbiota recovered from periodontitis specimen. Although their pathogenetic role in periodontal lesions remains unclear, their presence in the oral cavity should be recognized as a potential reservoir for development and spread of antibiotic resistance in light of antibiotic usage in oral infections.

## 1. Introduction

The periodontal microbiota is ecologically diverse and may facilitate colonialization of species that are not usually part of the oral microbiota [1]. In this study, we reasoned that dysbiosis associated with periodontitis could foster colonization and growth of atypical pathogens and could thereby turn into an important reservoir for survival and spread of antibiotic resistant pathogens.

*Enterococci*, for example, are considered as transient constituent components of the oral microbiome and it is well known that they can cause or contribute to a variety of oral and systemic infections including urinary tract, blood stream, and wound infections, or endocarditis [2,3,4,5]. They have also been recognized as nosocomial pathogens, mainly due to the increasing emergence of antimicrobial resistance phenotypes [6,7] and their capacity to form biofilms. Well in line with these characteristics, they are also regularly recovered from periodontal pockets in diseased patients [8,9,10,11]. Thus, opportunistic microbiota, such as *Enterococci* and key periodontal pathogens, are conjointly recovered from periodontal pockets [12,13], suggesting a role for oral commensals in periodontal tissue destruction.

By contrast, *Enterobacteriaceae*, such as *E. coli*, are primarily localized in the intestinal tract and are neither usually detected in the oral cavity nor considered oral microbiota [14]. Their presence may vary depending on age and environmental exposure [15]. Similarly, to *Pseudomonas aeruginosa* they gain access to the pharynx and oral cavity from external sources or transiently colonize the upper respiratory tract after displacement following regurgitation or resuscitation measures [16]. However, their presence in the oral cavity has not been thoroughly investigated.

Within the high heterogeneity of the human microbiota, *Candida* species are commensal microorganisms in healthy individuals, but can become pathogenic following changes in the host environment and dysbiosis [17]. Overgrowth on mucosal surfaces is commonly observed in immune compromised conditions or following antibiotic treatments [18,19,20].

As noted above, enterococcal species can produce biofilms, which promotes tolerance to antimicrobials and hinders penetration of these substances and, additionally, enterococcal species, especially *E. faecium*, accumulate genetic antibiotic resistance features [10,21]. Similarly, *Enterobacteriaceae* represent an important reservoir for antibiotic resistance elements that can become a dangerous endogenous source for infection.

Antimicrobial resistance (AMR) has become a major health problem with a magnitude at least as large as major diseases such as HIV and malaria, potentially even more relevant [22], mainly related to the indiscriminate usage of antibiotics. As a consequence, there is an urgent need to search for new strategies to combat antimicrobial resistance.

Thus, the aim of this study was to investigate the prevalence of oral and enteric opportunistic pathogens, e.g., *Enterobacteriaceae*, *Enterococci*, and *Candida albicans* in periodontal pockets. We further analyzed the antibiotic susceptibility trends of selected bacterial species (*Enterococcus* spp., *Enterobacter* spp., *Klebsiella* spp., and *Serratia* spp.) in periodontal lesions of a large patient population over a time period of eight years.

## 2. Results

Subgingival bacterial isolates were recovered from 16,612 patients diagnosed with periodontitis in Germany between 2008 and 2015. Information on medical status, prior history of antibiotic usage or systematic collection of data on the clinical course were not available. Patients ranged from 11 to 89 years old with an average of 51.8 years. The presence of classical periodontal pathogens was confirmed in a previous report [23].

Here, we were interested in the prevalence of non-periodontal species in samples of the subgingival biofilm, such as Gram-negative enteric rods and *Enterococci as* well as co-infections with *Candida* spp.

The prevalence (number of positive samples per bacterial species in the whole patient population) was calculated and the proportional representation of the isolated bacterial species in the total patient population (percentage (%) of patient population harboring the bacterial species) were grouped by year (Table 1a).

In all specimens investigated from the years 2008 to 2015, *Enterococcus* spp. Was present in 5.6% of the population, *E. cloacae* in 2.5%, *K. oxytoca* in 3.6%, *K. pneumonia* in 1.1%, *S. marcescens* in 2.8%, and *S.*
*liquefaciens* in 1.5%. *Candida albicans* was consistently detected in 21.8% of the cases over the whole course of the surveillance period. Dual co-infections of *Candida* with the selected *Enterobacteriaceae* and *Enterococci* ranged from 0.9 to 6.5% (Table 1b).

Not surprisingly, we observed high resistance patterns to AML, AMC, DO, and CIP because species harboring intrinsic resistance to aminopenicillins (*Enterobacter* spp., *Klebsiella* spp., *Serratia* spp.) were predominant. *Notably, Serratia* spp. Isolates were consistently highly resistant to AML (*S. marcescens:* 100%*; S.*
*liquefaciens:* 93.4%) (Table 2a,b). *S. marcescens* showed in vitro tolerance to DO of 37.7% and CIP of 1.3% and *S.*
*liquefaciens* to AMC of 38.2%, DO of 4.2%, and CIP of 1.2%. For *Serratia* spp., there were trends of decreasing resistance to DO in *S. marcescens* and to AMC in *S. liquefaciens.* The latter showed an increasing resistance trend to AML (Table 2a,b).

*Klebsiella* spp. Harboring intrinsic resistance to aminopenicillins were sensitive to aminopenicillins in the presence of beta-lactamase inhibitors (AMC) in most isolates. Notably, there was a trend towards decreasing resistance profiles to AMC for *K. oxytoca*. In vitro tolerance to DO remained below 20% (year 2014/*K. pneumoniae*) and below 12.5% to CIP (year 2012). Results for *Klebsiella* spp. Are summarized in Table 3a,b.

*Enterobacter cloacae* being naturally resistant to AML, the majority of isolates (up to 96%, year 2008) remained resistant to AMC. Overall, about 20.4% of the isolates were found to be tolerant to DO, with a trend to decreasing resistance profiles over time. Up to 5.9% of the isolates were resistant to CIP (Table 4).

All *Enterococcus* isolates showed substantial resistance rates to DO, 49.6% on average, and to CIP (28.7%). There were trends of decreasing resistance to CIP (*p* < 0.05). Conversely, there was an increase in resistance profiles for DO (*p* < 0.05). All isolates were sensitive to AML and AMC. The results are summarized in Table 5.

## 3. Discussion

This retrospective surveillance study aimed to examine bacterial subgingival samples from periodontitis patients for microbiological superinfection in the context of the global increase in AMR, a problem whose magnitude is at least as large as major diseases [22]. To the best of our knowledge, this is (so far) the largest study to report on the prevalence of oral/periodontal *Enterobacteriaceae/Enterococci* and in vitro antibiotic susceptibility trends in periodontitis patients. Our study confirmed that subgingival plaque samples from German periodontitis patients did not only contain species of the so called “Socransky-complexes” [23] but also confirmed the previously reported presence of opportunistic commensal pathogens [10,11,24,25,26]. The data revealed a consistent presence of *Enterobacteriaceae* and *Enterococci* in about 5%—as well as the presence of *Candida albicans* in about 22%—of the patients.

Several studies have addressed antibiotic susceptibility of *Enterococci* or *Enterobacteriaceae* in periodontal infections but comparison of our data with studies from Norway, US, Sri Lanka, India, Brazil, or Iran revealed that the numbers of patients included in the other studies were lower, e.g., with a study size of 23 [27], 30 [13], 70 [21], 70 [28], 169 [8], 205 [11], 305 [10], 400 [29], or 564 patients [30].

Most *Enterobacteriaceae* isolates collected from 16,000 German periodontitis patients displayed resistance to the antibiotics tested suggesting at least partial inefficacy of adjunctive antibiotic therapy with amoxicillin or doxycycline frequently used in periodontitis, if *Enterobacteriaceae* actively contributed to infection. However, it remains to be clarified whether the presence of *Enterobacteriaceae* in periodontal pockets is indicative of infection, contributes to disease course, or rather serves as a microbiological placeholder following antibiotic treatment and destruction of oral microbiota.

*Serratia* spp. And *Klebsiella* spp. Were recovered from the subgingival microbiota of Sri Lankan tea laborers at an average of 3% [13], as well as from other investigators [24,25,31,32]. A very recent study from Brazil showed a frequency of detection of *K. oxytoca* of 5%, *K. pneumonia* of 9%, *S. marcescens* of 3%, *liquefaciens* of 2%, or *Enterobacter cloacae* of 14% in periodontitis patients [11].

Overall, these studies are very heterogeneous because species were presented using different methodologies. Furthermore, differences may be explained by subject number, local epidemiology, microbiological methodology, as well as massive uncontrolled use of antibiotics. Our study provides data on the presence or absence of individual species in each patient and prevalence of enteric periodontitis microbiota within a German patient cohort. Furthermore, we were able to show that *Serratia* spp., *Klebsiella* spp., or *Enterobacter* spp. Are consistently present in periodontitis patients over a long surveillance observation period although in low frequencies.

Gonçalves et al. [28] found the majority of the enteric rods resistant to AMC (81.25%). A more recent study by [11] also found *S. marcescens*, *E. cloacae* with high resistance rates to AMC >40%. This is in agreement to our findings of high resistance rates in *S. marcescens* and *E. cloacae* isolates, which is expected and based on their natural antibiotic resistance profiles [33]. In contrast to our finding that *Serratia* spp., *Klebsiella* spp., and *Enterobacter* spp. Showed resistance rates to CIP to some extent, all strains from Rio de Janeiro appeared susceptible to CIP. Among the selected *Enterobacteriaceae* in Brazil (*Serratia* spp., *Klebsiella* spp., and *Enterobacter* spp. Included), 25% of isolates showed resistance to DO [28] in comparison to a range between 37.7% (*S. marcescens*) and 4.2% (*S. liquefaciens*) tested in our German patient cohort. Here, in our study, high resistance rates to amoxicillin with ß-lactamase-inhibitor clavulanic acid were observed in *E. cloacae* (79.1%). Generally, to aminopenicillins with ß-lactamase inhibitors, such as clavulanic acid (AMC), isolates exhibited lower resistance rates than to aminopenicillin alone, both in Germany and in Brazil.

In the United States, *Enterobacter* isolates, the second most common carbapenem-resistant *Enterobacteriaceae species*, increasingly contribute to the spread of carbapenem-resistant infections [34]. Resistance to these last-resort antibiotics and the emergence of multidrug resistance has led to an increased interest in these organisms because *Enterobacter cloacae* complexes (ECC) are common nosocomial pathogens capable of producing a wide variety of infections and septicemia [8,35]. Together with intrinsic ß-lactam resistance, members of the ECC exhibited a unique ability to acquire genes encoding resistance to multiple classes of antibiotics and contribute to global expansion of carbapenem-resistant *E. cloacae* complexes, and are becoming a diversifying threat [36].

The most common *Enterococci* species studied is *E. faecalis*, which has previously been recovered from periodontal pockets in 1% to 52% of periodontitis patients [8,9,21,37]. In our study, about 5% *Enterococci* spp. Positive subjects compared well to 4% in the United Stated [29]. With 9.8% *Enterococci* spp. These numbers from Brazil, derived from 305 periodontitis patients, appeared to be twice as high [10]. It has been observed that there is an increasing degree of carriage in the adults and elderly [1].

*Enterococci* are generally considered as transient oral bacteria. However, it is noteworthy that some of the above studies reported a relatively high intra-oral prevalence of *E. faecalis*. In agreement with the previously published studies, our study population most likely mainly harbored *E. faecalis*, which is generally less prone to resistance than *E. faecium*. This is supported by the observation that the enterococcal isolates were AML and AMC susceptible, which is not observed in *E. faecium* [38]. It can only be speculated that differences in patient selection (nutrition, health status, genetics, wearing dentures, use of mouth rinses, and medication including antibiotics) and/or possible virulence of epidemic strains could be the reason for these observations.

Our results showed that co-cultivation of *Candida albicans* with enteric isolates ranged from 0.9 to 6.5% (Table 1b). The ecological diversity of the periodontal micro-environment obviously provides suitable conditions for the colonization of these species not usually considered members of the oral microbiota. *Enterobacteria* and *Candida albicans* were related to periodontal inflammation and tissue destruction at the patient and/or site levels investigated [17,18,24]. Many oral mucosal infections contain a mixture of opportunistic pathogens, such as *Enterobacteriaceae, Pseudomonas* spp., *Enterococci*, and *Candida* spp., and can function as sources of periodontal pocket colonization. In these polymicrobial communities, an extracellular matrix may cover and protect biofilm cells from the surrounding environment. Furthermore, microorganisms secrete quorum-sensing molecules that control biological activities/behaviors and play a role in fungal pathogenicity [39]. Fungal infections are classical oral opportunistic infections caused by either systemic impairment of the host (cytotoxic drugs, human immunodeficiency virus infection) or local factors [25]. In general, fungi constitute a relatively small percentage (<0.1%) of the oral microbiome and more than 30% of the of the oral mycobiome members have not yet been cultivated or taxonomically classified. *C. albicans* continues to be the major fungal species associated with infection [17,40]. Numerous physical, signaling, and metabolic interactions may occur between oral bacteria and *Candida*, which can lead to both synergistic and antagonistic disease outcomes. Recent microbiome studies showed that the number of fungal species is reduced in oral candidiasis, compatible with oral microbial dysbiosis [17].

Subgingival *Enterococci*, especially *E. faecalis*, have been described as resistant to routine antimicrobial agents in high proportions [27,29,41]. Furthermore, it has been recently shown that *Enterococci* adapt by acquiring transferable antimicrobial resistance and are likely to be a reservoir for diverse mobile genetic elements [28]. The most common *Enterococci species*, *E. faecalis* and *E. faecium*, can produce biofilms with a significant association with resistance to drug penetration [21]. In fact, previous studies have shown that the oral cavity can constitute a reservoir for virulent *E. faecalis* strains possessing antibiotic resistance traits and at the same time distinct biofilm formation capabilities facilitating exchange of genetic material [42]. This feature may enhance pathogenicity and aggravate the disease course.

*Enterococci*-isolates from the present study displayed considerable resistance to DO and CIP. About half of the *Enterococci*-isolates (49.6%) were resistant to DO, overall, 28.7% of *Enterococci*-isolates were found to be resistant to CIP. All isolates were sensitive to aminopenicillins (AML, AMC). This compares well to low AML-resistance (4.3%), and high DO-resistance in India (53.8%) [21] or 53.2% DO-resistance in Brazil [1], respectively, whereas CIP-resistance appeared to be significantly higher in our cohort than in India, with 8.7%. Other investigations found *Enterococci* isolates resistant to DO, susceptible to Ampicillin, AMC, and CIP [1,41,43,44]. The latter antibiotic was shown to correlate with CIP usage among hospitalized patients, which can result in the selection of CIP resistant *Enterococci* strains, including VRE [44,45].

While antibiotic resistances among *Enterococci* in periodontal infections have been previously investigated to some extent, antimicrobial resistances in enteric rods conjointly recovered from periodontal pockets are sparsely investigated. Our study with an observation period of over 8 years, revealed oral/periodontal carriage of these organisms and colonization profiles of *Enterococci* in about 5% of patients. In agreement with a study from the US, we suspect that *Enterococci* spp. probably belong to the habitat of the periodontal pockets. While Gram negative enteric rods and *Enterococci* are classical opportunists in oral infections, little is known about their contributions to the progression of periodontal infection or in the failure of periodontal anti-infective therapy [46].

Trend analysis showed that over the study period between 2008 and 2015, resistance of *Enterococci*-isolates to DO increased, while resistance to CIP decreased. A decrease in the resistance trend was also observed for *E. cloacae* to DO. Results in the trend analysis show fluctuations in antibiotic susceptibilities among enteric rods and underline the already established understanding that antibiotic susceptibility among bacterial species change over time and may exhibit geographical variation [29]. These fluctuations do not provide evidence for inefficiency of the currently recommended prescription regimens with antibiotics in general because the occurrence of *Enterococci* or *Enterobacteria* is low. However, matters of concern are the presence of *Enterobacteriaceae* resistant to virtually all antimicrobials currently used in clinical practice. These organisms are well adapted to survive in their habitat and can become the dominant flora under antibiotic pressure, predisposing the severely ill and immunocompromised patient to invasive infections [20,47,48].

Amoxicillin with or without ß-lactamase inhibitor is still commonly used in periodontal anti-infective therapy [49]. Here, we found increasing resistance profiles to AML in *S. liquefaciens* together with species of high intrinsic resistance (*S. marcescens*, *Klebsiella* spp., *E. cloacae*). Consequently, amoxicillin as an adjunct treatment in periodontal infections, harvesting heavy *Enterobacteriaceae* colonization, poses a risk of post-treatment emergence of superinfection by multi-resistant species and clinical treatment failure [50]. The occurrence of enteric opportunists with resistance to multiple antibiotics regularly used in systemic infections, as presented in this study, emphasizes the magnitude of possible dissemination of infections from periodontal pockets to other sites of the body, particularly in immuno-compromised or hospitalized patients.

In our study, lack of information on prior antibiotic prescriptions does not allow conclusions on resistance development in the individual patient/cohort. However, available studies [45,51,52] addressing this important question found a relationship between antimicrobial usage and incident resistant *Enterococci* colonization at the individual patient/cohort level. McKinnell et al. [52] concluded that the risk differs between individual antibiotic agents and supports the significance of antimicrobial stewardship. As with most large observational reports, the present study has limitations. The selected study patient pool originated from private dental practices and, therefore, does only reflect a specific cross-section of periodontitis patients in Germany. Comparability and generalizability of the present results to other countries is limited.

As discussed previously, the database used for this study has some inherent weaknesses, such as the lack of clinical data and of an examiner calibration with regard to patient inclusion into the study [23]. Thus, an analysis or interpretation with regard to the clinical significance of the findings is not feasible. In addition, in vitro antibiotic susceptibility testing of individual isolates is not directly applicable to in vivo drug effectiveness, especially in a subgingival biofilm environment. It is therefore not possible to extrapolate the in vitro results to the individual patient setting. Notably, periodontitis is associated with biofilm formation and a high propensity among the species recovered in this study to form biofilms (e.g., *E. cloacae*, *Klebsiella* spp., *S. marcescens*, in addition to *Enterococci*). However, it is unclear whether biofilm formation merely synergizes with present antibiotic resistance or actively supports the development of antibiotic resistance. Current publications indicate that species-specific effects might strongly influence biofilm formation and associated susceptibility to antibiotics [53,54,55]. Therefore, in future prospective studies, it may be necessary to develop in vitro systems that reflect the impact of biofilm formation on antibiotic penetration and susceptibility. Nevertheless, it is commonly accepted that antimicrobial agents failing to inhibit bacterial growth in vitro, are most likely ineffective antimicrobial agents in vivo as well.

Periodontitis is the most common chronic inflammatory non-communicable disease of humans. According to data originating from the Global Burden of Disease database, 1.1 billion cases of severe periodontitis were prevalent globally in 2019, and an 8.44% (95% confidence interval—CI—6.62%–10.59%) increase in the age-standardized prevalence rate of severe periodontitis was observed between 1990 and 2019 [56]. In regard to the large population affected, we have to use antimicrobials responsibly, even though they are known to be effective as adjuncts to mechanical debridement in the treatment of periodontitis. This involves promoting antibiotic stewardship measures that acknowledge the individual’s need for appropriate treatment and the longer-term societal need for sustained access to effective therapy. With this in mind, the European Federation of Periodontology has published periodontal treatment guidelines that call for prudent use of adjunctive systemic antibiotics [57].

Additionally, clinicians should be aware that periodontal pockets may (so far) serve as neglected reservoirs of virulent and resistant *Enterococci* and *Enterobacteriaceae*. Hematogenic spread and translocation of pathogens to other areas of the body occur and may cause serious diseases, such as brain abscesses, lung infections, endocarditis, and soft tissue infections. Consequently, the surveillance of prevalence and susceptibility of the subgingival microbiota may become relevant in the prevention and treatment of infections with endogenous pathogens.

The results of the present study, together with a recent investigation [23], illustrate that the variabilities of antibiotic susceptibility profiles among subgingival pathogens in a mixed facultative–anaerobe biofilm environment in periodontal pockets complicate the selection and administration of adequate antibiotic regimens. The findings highlight the urgency to establish a monitoring system for antibiotic resistance and consumption of antibiotics, along with the critical need to set up strategies for prudent administration of antibiotics in periodontal treatment. Prospective clinical and surveillance studies should be performed to establish efficacy of antibiotic regimens and should be supported by molecular analysis using next generation sequencing methods. Future molecular studies are expected to complement the present findings by providing insight into antibiotic resistance-linked alterations in the microbiome composition, accumulation of genetic resistance elements, and metagenomic patterns reflecting dysbiosis and tolerance to antibiotics.

## 4. Materials and Methods

### 4.1. Setting and Patients

We performed a retrospective analysis of microbiological data collected from 2008 to 2015 in a laboratory specialized in oral microbiology. The data were obtained from routine microbiological examinations of cultures of subgingival plaque samples, harvested from deep inflamed periodontal pockets in patients (diagnosed by individual clinicians) with moderate to advanced periodontitis [58], in 160 German dental offices. The samples were collected prior to treatment from a total of 16,612 patients. The study was approved by the ethics committee of the University of Bonn (no. 370/19). Results of the analyses were summarized and processed anonymously.

### 4.2. Microbiological Sampling and Transport

Subgingival plaque specimens were procured by the diagnosing dentists prior to treatment following a standardized sampling protocol. Methods were reported in a previous publication [23]. Briefly, one to five sterile absorbent paper points (Hain Lifescience GmbH, Nehren, Germany) were introduced into up to 5 different deep pockets with a depth of ≥6 mm for 20 s. After removal, all paper points per patient were pooled into one glass vial containing AMIES transport medium (Mast Diagnostica, Reinfeld/Stormarn, Germany). Altogether, one pooled sample per patient was available for the microbiological examination.

### 4.3. Microbiological Cultures and Species Identification

Laboratory procedures were performed according to established quality standards [59,60,61]. In the laboratory, paper points were transferred into a 2 mL pre-reduced thioglycolate suspension (Oxoid™/Fisher Scientific, Munich, Germany) and microorganisms were mechanically dispersed by vortexing. Serial, 10-fold dilutions were inoculated on Columbia sheep blood agar, China lactose blue agar, and Kimmig agar (Oxoid™/Fisher Scientific) followed by incubation at 37 °C for 24 h in aerobic conditions. Recovered patient isolates were identified on a species level and subjected to antimicrobial susceptibility testing, but not quantified.

### 4.4. In Vitro Antibiotic Susceptibility Testing (AST)

Patient isolates of *Serratia* spp., *Klebsiella* spp., *Enterobacter cloacae*, and *Enterococcus* spp. were subjected to antimicrobial susceptibility testing (AST). After subculturing, bacterial suspensions were prepared from pure cultures and adjusted to a 0.5 McFarland by turbidity measurement in saline solution. Moreover, 0.1 mL aliquots were inoculated onto Mueller Hinton Agar (Oxoid™/Fisher Scientific, Munich, Germany).

Antibiotic discs (Oxoid™/Fisher Scientific, Munich, Germany) for in vitro susceptibility testing contained 10 μg amoxicillin (AML), 30 μg amoxicillin/clavulanic acid (AMC 2:1), 30 μg doxycycline (DO), or 5 μg ciprofloxacin (CIP) (Oxoid™/Fisher Scientific, Munich, Germany). After incubation at 37 °C for 24 h, the diameter of the growth inhibition zones was measured and isolates were graded as sensitive or resistant according to the individual most recent available tables available for interpretation of zone diameters (https://www.eucast.org (accessed on 4 November 2021)).

### 4.5. Statistical Analysis

Analyses of the total patient population were based on the data obtained on the recovery of bacterial species on the patient (sample) level as well as for the occurrence of growth inhibition by antibiotics of selected species. For each combination of bacterial specimen and antimicrobial substance, the temporal evolution of antibiotic resistance from 2008 to 2015 was analyzed using linear logistic regression, modeling the time dependency of the rates of non-susceptible patient isolates over the years, with time as the only covariate. Data analysis was performed using SAS^®^ (SAS Institute Inc., Cary, NC, USA).

## 5. Conclusions

The present study within its limitations—such as the retrospective design, lack of bacterial quantification, and limited clinical information—confirmed the presence of *Enterobacteriaceae, Enterococci*, and *Candida* in deep periodontal pockets of German periodontitis patients over an observation period of 8 years.

*Enterobacter cloacae*, *Serratia marcescens*, and *Klebsiella pneumoniae* displayed the expected natural resistance to amoxicillin. Trends of increasing resistance were found to AML in *Serratia liquefaciens* and to DO in *Enterococcus* spp. Trends of decreasing resistance to AMC were observed in *Serratia liquefaciens* and *Klebsiella oxytoca*, to DO in *Serratia marcescens, liquefaciens*, and *Enterobacter cloacae*, and to CIP in *Enterococcus* spp.

Taken together, our data confirm that periodontitis is a multiple species infection, which argues for a (co-)pathogenic role of *Enterobacteriaceae* as well as *Candida* spp., in addition to the classical periodontal pathogens. All of these species are known biofilm formers, which could contribute to increased tolerance to antibiotics. Furthermore, a predominance of species with resistance to aminopenicillins, doxycycline, and quinolones might result from previous exposure to oral antibiotics.

In the future, novel next generation sequencing methods could provide information on resistance profiles of bacterial biofilms containing multiple species and could thereby better reflect susceptibility profiles guiding antibiotic regimen selections.

## Figures and Tables

**Table 1 antibiotics-11-00385-t001:** (**a**) Prevalence of periodontitis patients (*n* = 16,612) harboring the targeted commensal bacteria or candida species in Germany. (**b**) Prevalence of periodontitis patients (*n* = 16,612) harboring the targeted commensal bacteria with *Candida albicans* co-infections.

**(a)**									
**Year**	**2008**	**2009**	**2010**	**2011**	**2012**	**2013**	**2014**	**2015**	**2008–2015**
Number of Patients	2692	1984	1903	2104	1942	1808	2014	2165	16,612
samples positive with species (%)									
*S. marcescens*	2.6	3.3	2.6	2.6	2.3	2.9	3.3	3.1	2.8
*S. liquefaciens*	0.5	1.3	1.4	2.1	2.1	1.3	1.4	1.8	1.5
*K. pneumonia*	0.8	0.7	1.7	1.4	0.4	1.2	1.2	1.5	1.1
*K. oxytoca*	2.8	4.7	3.6	4.6	3.8	3.2	2.9	3.5	3.6
*Enterobacter cloacae*	2.0	2.1	2.5	3.3	2.6	2.7	2.3	2.5	2.5
*Enterococcus* spp.	3.5	5.1	5.9	5.4	6.9	6.6	6.7	5.8	5.6
*Candida albicans*	14.7	22.8	23.0	22.4	22.9	20.9	25.9	24.0	21.8
**(b)**									
**Year**	**2008**	**2009**	**2010**	**2011**	**2012**	**2013**	**2014**	**2015**	**2008–2015**
Number of Patients	395	453	438	472	444	378	521	520	3621
samples positive with species *Candida albicans* co-infections (%)									
*S. marcescens*	4.6	1.3	3.2	0.2	2.3	2.1	2.1	1.7	2.1
*S. liquefaciens*	0.5	1.1	0.9	0.8	1.4	1.1	1.0	0.8	0.9
*K. pneumonia*	1.3	0.4	2.3	1.3	0.9	1.3	1.2	1.3	1.2
*K. oxytoca*	3.8	3.3	2.5	4.0	2.7	2.6	3.1	1.7	2.9
*Enterobacter cloacae*	2.0	1.5	2.3	3.2	3.4	2.6	1.7	1.0	2.2
*Enterococcus* spp.	7.8	3.3	6.4	4.9	8.1	9.3	6.0	7.1	6.5

**Table 2 antibiotics-11-00385-t002:** (**a**) Temporal evolution (2008 to 2015) of the occurrence of antibiotic resistances to amoxicillin (AML), amoxicillin/clavulanic acid (AMC), doxycycline (DO), ciprofloxacin (CIP) was analyzed using logistic regression methods. Trend in the occurrence of antibiotic resistances (%) in *Serratia marcescens.* (**b**) Temporal evolution (2008 to 2015) of the occurrence of antibiotic resistances to amoxicillin (AML), amoxicillin/clavulanic acid (AMC), doxycycline (DO), ciprofloxacin (CIP) was analyzed using logistic regression methods. Trend in the occurrence of antibiotic resistances (%) in *Serratia liquefaciens*.

**(a)**														
**Year**	**2008**	**2009**	**2010**	**2011**	**2012**	**2013**	**2014**	**2015**	**2008–2015**	**Logistic Regression Analyses**				
patients (*n*)	70	66	49	55	45	53	67	67	472	Odds Ratio	Confidence interval 95%		estimate	*p*-value
* *Serratia marcescens* resistant to											lower	upper		
* AML	100	100	100	100	100	100	100	100	100.0	*n*/a				
AMC	87.1	89.4	81.6	83.6	80.0	69.8	86.6	83.6	83.3	0.940	1.040	0.850	−0.062	0.229
DO	44.3	45.5	30.6	49.1	40.0	22.6	26.9	40.3	37.7	0.925	1.000	0.856	−0.078	0.049
CIP	4.3	0.0	0.0	0.0	4.4	0.0	0.0	1.5	1.3	0.837	1.193	0.588	−0.177	0.325
**(b)**														
**Year**	**2008**	**2009**	**2010**	**2011**	**2012**	**2013**	**2014**	**2015**	**2008–2015**	**Logistic Regression Analyses**				
patients (*n*)	14	26	27	42	39	24	29	40	241	Odds Ratio	Confidence interval 95%		estimate (year)	*p*-value
*Serratia liquefaciens* resistant to											lower	upper		
AML	92.9	80.8	77.8	95.2	100	100	100	95.0	93.4	1.499	1.976	1.137	0.405	0.004
AMC	85.7	73.1	44.4	33.3	10.3	16.7	34.5	42.5	38.2	0.797	0.905	0.701	−0.227	0.001
Do	7.1	7.7	0	14.3	0.0	0.0	0.0	2.5	4.2	0.754	1.038	0.548	−0.282	0.083
CIP	0.0	7.7	0.0	0.0	0.0	0.0	0.0	2.5	1.2	0.819	1.422	0.472	−0.199	0.479

* Species harboring intrinsic resistance to aminopenicillins; *n*/a: not applicable.

**Table 3 antibiotics-11-00385-t003:** (**a**) Temporal evolution (2008 to 2015) of the occurrence of antibiotic resistances to amoxicillin (AML), amoxicillin/clavulanic acid (AMC), doxycycline (DO), ciprofloxacin (CIP) was analyzed using logistic regression methods. Trend in the occurrence of antibiotic resistances (%) in * *Klebsiella pneumoniae.* (**b**) Temporal evolution (2008 to 2015) of the occurrence of antibiotic resistances to amoxicillin (AML), amoxicillin/clavulanic acid (AMC), doxycycline (DO), ciprofloxacin (CIP) was analyzed using logistic regression methods. Trend in the occurrence of antibiotic resistances (%) in * *Klebsiella oxytoca*.

**(a)**														
**Year**	**2008**	**2009**	**2010**	**2011**	**2012**	**2013**	**2014**	**2015**	**2008–2015**	**Logistic Regression Analyses**				
patients (*n*)	21	14	32	30	8	22	25	32	184	Odds Ratio	Confidence interval 95%		estimate (year)	*p*-value
** Klebsiella pneumonia* resistant to											lower	upper		
AML	100	100	100	100	100	100	100	100	100	*n*/a				
AMC	9.5	14.3	0.0	0.0	12.5	0.0	4.0	9.4	4.9	0.990	1.317	0.745	0.010	0.947
DO	4.8	14.3	3.1	6.7	12.5	9.1	20.0	9.4	9.2	1.153	1.436	0.925	0.142	0.206
CIP	4.8	7.1	0.0	0.0	12.5	0.0	0.0	0.0	1.6	0.635	1.208	0.334	0.455	0.166
**(b)**														
**Year**	**2008**	**2009**	**2010**	**2011**	**2012**	**2013**	**2014**	**2015**	**2008–2015**	**Logistic Regression Analyses**				
patients (*n*)	75	94	68	96	74	58	59	76	600	Odds Ratio	Confidence interval 95%		estimate (year)	*p*-value
* *Klebsiella oxytoca* resistant to											lower	upper		
* AML	100	100	100	100	100	100	100	100	100	*n*/a				
AMC	21.3	3.2	0.0	7.3	1.4	3.5	1.7	6.6	5.8	0.789	0.933	0.667	−0.237	0.006
DO	5.3	3.2	10.3	9.4	6.8	3.5	3.4	5.3	6.0	0.970	1.125	0.835	−0.031	0.684
CIP	5.3	0.0	2.9	0.0	1.4	3.5	0.0	0.0	1.5	0.746	1.050	0.530	−0.293	0.093

* Species harboring intrinsic resistance to aminopenicillins; *n*/a: not applicable.

**Table 4 antibiotics-11-00385-t004:** Temporal evolution (2008 to 2015) of the occurrence of antibiotic resistances to amoxicillin (AML), amoxicillin/clavulanic acid (AMC), doxycycline (DO), ciprofloxacin (CIP) was analyzed using logistic regression methods. Trend in the occurrence of antibiotic resistances (%) in * *Enterobacter cloacae*.

Year	2008	2009	2010	2011	2012	2013	2014	2015	2008–2015	Logistic Regression Analyses				
patients (*n*)	54	41	47	69	51	48	47	54	411	Odds Ratio	Confidence interval 95%		estimate (year)	*p*-value
** Enterobacter cloacae* resistant to											lower	upper		
AML	100	100	100	100	100	100	100	100	100	*n*/a				
AMC	96.3	92.7	59.6	72.5	74.5	62.5	87.2	88.9	79.1	0.957	1.063	0.861	−0.044	0.410
DO	24.1	46.3	14.9	24.6	9.8	12.5	10.6	22.2	20.4	0.869	0.968	0.780	−0.140	0.011
CIP	0.0	0.0	4.3	1.5	5.9	0.0	2.1	1.9	2.0	1.101	1.505	0.805	0.096	0.546

* Species harboring intrinsic resistance to aminopenicillins; *n*/a: not applicable.

**Table 5 antibiotics-11-00385-t005:** Temporal evolution (2008 to 2015) of the occurrence of antibiotic resistances to amoxicillin (AML), amoxicillin/clavulanic acid (AMC), doxycycline (DO), ciprofloxacin (CIP) was analyzed using logistic regression methods. Trend in the occurrence of antibiotic resistances (%) in *Enterococcus* spp.

Year	2008	2009	2010	2011	2012	2013	2014	2015	2008–2015	Logistic Regression Analyses				
patients (*n*)	93	102	112	113	134	119	135	126	934	Odds Ratio	Confidence interval 95%		estimate (year)	*p*-value
*Enterococcus* spp. Resistant to											lower	upper		
AML	0	0	0	0	0	0	0	0	0	*n*/a				
AMC	0	0	0	0	0	0	0	0	0	*n*/a				
DO	46.2	19.6	31.3	37.2	54.5	64.7	66.7	65.9	49.6	1.290	1.372	1.213	0.255	<0.0001
CIP	62.4	24.5	47.3	15.0	11.9	26.1	4.4	49.2	28.7	0.873	0.931	0.819	−0.136	<0.0001

*n*/a: not applicable.

## Data Availability

The data that support the findings of this study are available from the corresponding author upon reasonable request.

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
