# Peer review of "Prevalence and Antibiotic Susceptibility Trends of Selected Enterobacteriaceae, Enterococci, and Candida albicans in the Subgingival Microbiota of German Periodontitis Patients: A Retrospective Surveillance Study"

_antibiotics, 2022, doi:10.3390/antibiotics11030385_

Round 1
Reviewer 1 Report
Dear authors,
I really appreciate your work and efforts to carry out satisfactorily this study. In my opinion, your manuscript is generally well written and addresses a relevant topic, since antimicrobial resistance is growing every day. However, I have some suggestions that should be addressed in order to enhance the quality of your paper.
- Authors should take into account and even should indicate through the introduction section that the problem of antimicrobial resistance is mainly related to the indiscriminate usage of antibiotics. And, as a consequence, there is an urgent need to search for new strategies to combat antimicrobial resistance.
- Abstract:
- Please, mention which identification method was employed to identify bacterial isolates. In your case, culture-based approach.
- Line: 36: Serratia liquefasciens
- The font size and font type employed should be uniformed through all the section.
- Results:
- Table 1.a & 1.b : In order to unify the table and to reduce the size of bacterial names, I suggest to write them as: K. oxytoca , S. marcenses, etc.
- Line 101: Candida albicans
- Discussion:
- Line 175: S. liquefasciens
- Line 185: Gonçalves et al.
- The font size and font type employed should be uniformed through all the section.
- Material and methods:
- Line 374: Briefly,
- Line 392: The standard inoculum density of disk-diffusion method is 0.5 McFarland (https://www.eucast.org/ast_of_bacteria/disk_diffusion_methodology/). Please, justify why authors used 1.0 McFarland for the susceptibility assay.
- References:
- Scientific names must be written in italics, even in references. Please correct it. (for example: lines 470, 492).
Author Response
Dear Reviewer, we have carefully revised our manuscript according your comments. Revisions made to the manuscript are highlighted in yellow mark-ups in order to be easily viewed.
We wish to thank you for their valuable time and suggestions that have helped to improve our manuscript.

Reviewer 2 Report
The topic is relevant given the ever-present issue of antimicrobial resistance.
The large size of the study is a good point even if the transposition of the results is very uncertain. This study clearly appears to be a sub-analysis of the results of a previously published study.
Clinical relevance?
- No associated clinical parameters or information on patient medical history and individual factors that may explain transient oral carriage of enterobacteria and fungi
- Antibiotic choice in periodontal therapy is empirical and not based on the bacterial profile. The clinical value of studying the prevalence of specific bacteria is questionable.
- The Amoxicilline + metronidazole combination is the gold standard in periodontal treatment and has not been tested. Several molecules which have been tested have little or no relevance for periodontal treatment (quinolones, ciprofloxacin, etc.)
- The presence of these bacteria is not an argument in favor of their involvement in periodontal disease, its severity and progression (no correlation to possible clinical data). Moreover, it is accepted that periodontitis is not a "true infection" but a condition associated with dysbiosis. The relative proportion of microorganisms (or taxa) would therefore be more relevant than their mere detection.
Methodological issues?
For future studies (e.g. points to be addressed in discussion)
- The study of the expression of antimicrobial resistance genes by biomolecular techniques as a complement or alternative to susceptibility tests using cultured bacteria, which have many limitations and concerns.
- Evaluation of the susceptibility of the sample (multi-species) rather than of isolated strains
General question: Could the observed results be put into perspective with other data on the evolution of antibiotic consumption in the German population over the same period and/or data on the susceptibility of these bacteria when isolated from sites other than the oral cavity?
Minor comments: overall revision of the manuscript is recommended to detect some typos including "colonialization" instead of "colonization/colonization" and "chinolons" instead of "quinolones
Author Response
Dear reviewer, we have carefully revised our manuscript according your comments. Revisions made to the manuscript are highlighted in yellow mark-ups in order to be easily viewed. We wish to thank you for valuable time and suggestions that have helped to improve our manuscript. Yours sincerely Karin Jepsen

Reviewer 3 Report
Dears authors
In this retrospective surveillance study, he followed up the superinfection profiles associated with periodontitis in a large sample of patients. From 2008 to 2015, biofilm samples from deep periodontal pockets were collected from a total of 16,612 German adults diagnosed with periodontitis. (I think the number is reliable enough and more than correct for retrospective analysis.)
The presence of selected Enterobacteriaceae, Enterococci and Candida albicans was confirmed in nocturnal cultures. The antimicrobial susceptibility of these clinical isolates was tested by disc diffusion with antibiotics routinely used for the treatment of oral infections, e.g. amoxicillin (AML), amoxicillin / clavulanic acid (AMC), doxycycline (DO) and ciprofloxacin (CIP).
Among the Enterobacteriaceae, species presenting intrinsic resistance to AML (Enterobacter spp., Klebsiella spp., Serratia spp.) Were predominant. No susceptibility to AMC was observed for Serratia spp. and Enterobacter cloacae. Enterococcus spp. he only showed non-susceptibility to DO and CIP. Trends to increase resistance to AML have been found in Serratia liquefaciens and to DO in Enterococcus spp. Trend analysis showed decreasing resistance to AMC in Serratia liquefaciens and Klebsiella oxytoca; and DO in Serratia marcenses, liquefaciens and Enterobacter cloacae. This study confirms the low but consistent presence of Enterobacteriaceae and Enterococci in the subgingival microbiota recovered from the periodontitis sample.
Taking into account what is described above in the paper, some changes are necessary:
Analysis for paper partitions:
1 - Introduction: the contents and the drafting of the general part must be reformed to review the syntax of the topic
2- Discussion:
to investigate in consideration of the problem of antimicrobial resistance also of other species that cause periodontitis and phenotypic resistance to drugs and analysis of the biofilm formation capacity (lines: 312-314). Find out more about this by using and citing the following references:
PMID: 34572716 ; PMID: 32872324 ; PMID: 31702908
3 - Check the bibliographic entries throughout the text, some of which are non-compliant, review some entries in the references and necessarily insert those referred to in point 2 for the purpose of my acceptance.
4 - Review English grammar and in particular applied scientific English: in particular verb tenses and syntax in the discussion.
Author Response

(The authors gave the same response as above.)
